# Anytime Neural Network: a Versatile Trade-off Between Computation and Accuracy

**Hanzhang Hu, Martial Hebert & J. Andrew Bagnell**
Department of Computer Science
Carnegie Mellon University
{hanzhang, hebert, dbagnell}@cs.cmu.edu

**Debadeepta Dey**
Microsoft Research
dedey@microsoft.com

## Abstract

We present an approach for anytime predictions in deep neural networks (DNNs). For each test sample, an anytime predictor produces a coarse result quickly, and then continues to refine it until the test-time computational budget is depleted. Such predictors can address the growing computational problem of DNNs by automatically adjusting to varying test-time budgets. In this work, we study a *general* augmentation to feed-forward networks to form anytime neural networks (ANNs) via auxiliary predictions and losses. Specifically, we point out a blind-spot in recent studies in such ANNs: the importance of high final accuracy. In fact, we show on multiple recognition data-sets and architectures that by having near-optimal final predictions in small anytime models, we can effectively double the speed of large ones to reach corresponding accuracy level. We achieve such speed-up with simple weighting of anytime losses that oscillate during training. We also assemble a sequence of exponentially deepening ANNs, to achieve both theoretically and practically near-optimal anytime results at any budget, at the cost of a constant fraction of additional consumed budget.

## 1 Introduction

In recent years, the accuracy in visual recognition tasks has been greatly improved by increasingly complex convolutional neural networks, from AlexNet (Krizhevsky et al., 2012) and VGG (Simonyan & Zisserman, 2015), to ResNet (He et al., 2016), ResNeXt (Xie et al., 2017), and DenseNet (Huang et al., 2017b). However, the number of applications that require latency sensitive responses is growing rapidly. Furthermore, their test-time computational budget can often. E.g., autonomous vehicles require real-time object detection, but the required detection speed depends on the vehicle speed; web servers need to meet varying amount of data and user requests throughput through out a day. Thus, it can be difficult for such applications to choose between slow predictors with high accuracy and fast predictors with low accuracy. In many cases, this dilemma can be resolved by an **anytime predictor** (Horvitz, 1987; Boddy & Dean, 1989; Zilberstein, 1996), which, for each test sample, produces a fast and crude initial prediction and continues to refine it as budget allows, so that at any test-time budget, the anytime predictor has a valid result for the sample, and the more budget is spent, the better the prediction is.

In this work[1], we focus on the anytime prediction problem in neural networks. We follow the recent works (Lee et al., 2015; Xie & Tu, 2015; Zamir et al., 2017; Huang et al., 2017a) to append auxiliary predictions and losses in feed-forward networks for anytime predictions, and train them jointly end-to-end. However, we note that the existing methods all put only a small fraction of the total weightings to the final prediction, and as a result, large anytime models are often only as accurate as much smaller non-anytime models, because the accuracy gain is so costly in DNNs, as demonstrated in Fig. 1a. We address this problem with a novel and simple oscillating weightings of the losses, and will show in Sec. 3 that our small anytime models with near-optimal final predictions can effectively speed up two times large ones without them, on multiple data-sets, including ILSVRC (Russakovsky et al., 2015), and on multiple models, including the very recent Multi-Scale-DenseNets (MSDnets) (Huang et al., 2017a). Observing that the proposed training techniques lead to ANNs that are near-optimal in late predictions but are not as accurate in the early predictions, we

---

[1]For the full paper, see https://arxiv.org/abs/1708.06832

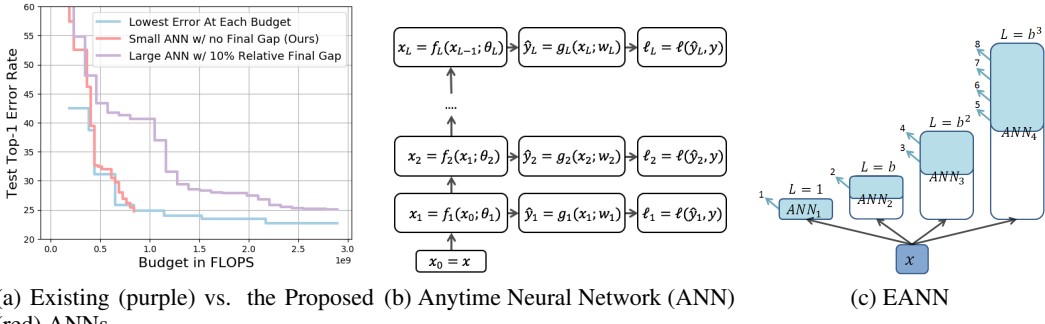

(a) Existing (purple) vs. the Proposed (b) Anytime Neural Network (ANN) (c) EANN
(red) ANNs

Figure 1: **(a)** By ensuring near-optimal final anytime predictions, our ANN (red) achieves the same error rates 2-5 times faster than a large existing one (purple). **(b)** ANN illustration. **(c)** EANN illustration.

assemble ANNs of exponentially increasing depths to dedicate early predictions to smaller networks, while only delaying large networks by a constant fraction of additional test-time budgets.

## 2 METHODS

As illustrated in Fig. 1b, given a sample $(x, y) \sim D$, the initial feature map $x_0$ is set to $x$, and the subsequent feature transformations $f_1, f_2, ..., f_L$ generate a sequence of intermediate features $x_i = f_i(x_{i-1}; \theta_i)$ for $i \geq 1$ using parameter $\theta_i$. Each feature map $x_i$ can then produce an auxiliary prediction $\hat{y}_i$ using a prediction layer $g_i$: $\hat{y}_i = g_i(x_i; w_i)$ with parameter $w_i$. Each auxiliary prediction $\hat{y}_i$ then incurs an expected loss $\ell_i := E_{(x,y)\sim D}[\ell(y, \hat{y}_i)]$. We call such an augmented network as an *Anytime Neural Network (ANN)*. Let the parameters of the full ANN be $\theta = (\theta_1, w_1, ..., \theta_L, w_L)$. The most common way to optimize these losses, $\ell_1, ..., \ell_L$, end-to-end is to optimize them in a weighted sum $\min_\theta \sum_{i=1}^{L} B_i \ell_i(\theta)$, where $\{B_i\}_i$ form the weight scheme for the losses.

**Alternating SIEVE weights.** Three experimental observations lead to our proposed SIEVE weight scheme. First, the existing weights, CONST (Lee et al., 2015; Xie & Tu, 2015; Huang et al., 2017a), and LINEAR (Zamir et al., 2017) both incur more than 10% relative increase in final test errors, which effectively slow down anytime models multiple times. Second, we found that a large weight can improve a neighborhood of losses thanks to the high correlation among neighboring losses. Finally, keeping a fixed weighting may lead to solutions where the sum of the gradients are zero, but the individual gradients are non-zero.

The proposed SIEVE scheme has half of the total weights in the final loss, so that the final gradient can outweigh other gradients when all loss gradients have equal two-norms. It also have uneven weights in early losses to let as many losses to be near large weights as possible. Formally for $L$ losses, we first add to $B_{\lfloor \frac{L}{2} \rceil}$ one unit of weight, where $\lfloor \bullet \rceil$ means rounding. We then add one unit to each $B_{\lfloor \frac{kL}{4} \rceil}$ for $k = 1, 2, 3$, and then to each $B_{\lfloor \frac{kL}{8} \rceil}$ for $k = 1, 2, ..., 7$, and so on, until all predictors have non-zero weights. We finally normalize $B_i$ so that $\sum_{i=1}^{L-1} B_i = 1$, and set $B_L = 1$. During each training iteration, we also sample proportional to the $B_i$ a layer $i$, and add temporarily to the total loss $B_L \ell_i$ so as to oscillate the weights to avoid spurious solutions. We call ANNs with alternating weights as alternating ANNs (AANNs). Though the proposed techinques are heuristics, they effectively speed up anytime models multiple times as shown in Sec. 3. We hope our experimental results can inspire, and set baselines for, future principled approaches.

**EANN.** Since AANNs put high weights in the final layer, they trade early accuracy for the late ones. We leverage this effect to improving early predictions of large ANNs: we propose to form a sequence of ANNs whose depths grow exponentially (**EANN**). By dedicating early predictions to small networks, EANN can achieve better early results. Furthermore, if the largest model has $L$ depths, we only compute $\log L$ small networks before the final one, and the total cost of the small networks is only a constant fraction of the final one. Hence, we only consume a constant fraction of additional test-time budget. Fig 1c shows how an EANN of the exponential base $b = 2$ works at test-time. The EANN sequentially computes the ANNs, and only outputs an anytime result if the current result is better than previous ones in validation. Formally, if we assume that each ANN has near-optimal results after $\frac{1}{b}$ of its layers, then we can prove that for any budget $B$, the EANN can

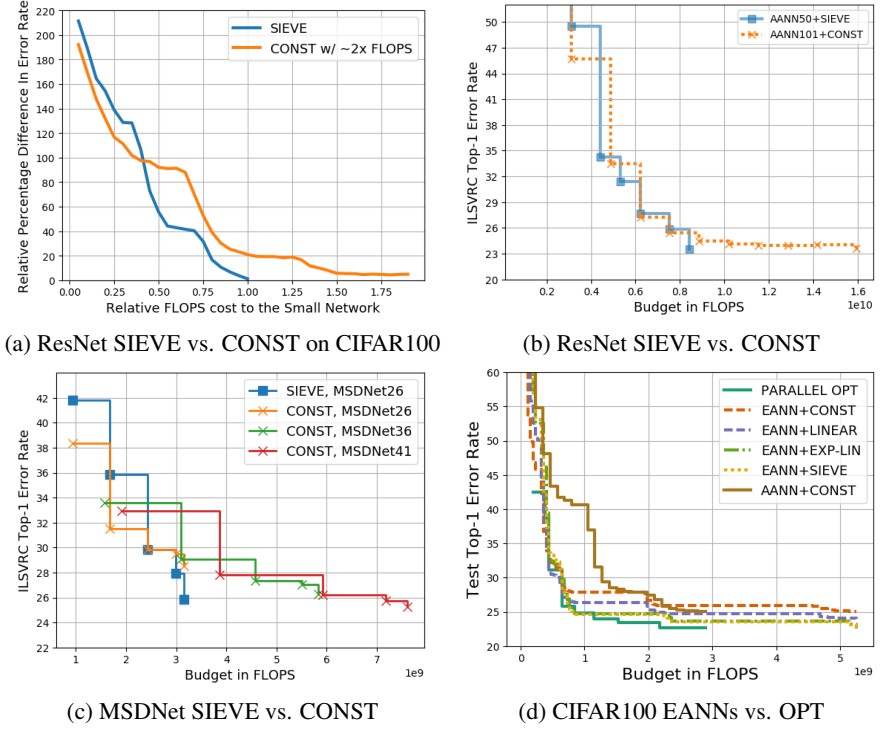

(a) ResNet SIEVE vs. CONST on CIFAR100

(b) ResNet SIEVE vs. CONST

(c) MSDNet SIEVE vs. CONST

(d) CIFAR100 EANNs vs. OPT

Figure 2: **(a,b)** On CIFAR100 and ILSVRC, ResNets using SIEVE achieves the same accuracy as those of twice the depths but using CONST, so that the small nets (blue) effectively speed up the large nets (orange). **(c)** Similar effects on ILSVRC with MSDNets: the small MSDNet26 with SIEVE (blue) is a better anytime predictor than MSDNet36 and MSDNet41 with CONST (green and red). **(d)** Using EANN, we greatly reduces the early error rates, at the cost of achieving the final predictions later.

achieve near-optimal predictions for budget $B$ after spending $C \times B$ total budgets. Furthemoe, for large $b$, $E_{B \sim uniform(1,L)}[C] \leq 1 - \frac{1}{2b} + \frac{1+\ln(b)}{b-1} \to 1$, and $\sup_B C = 2 + \frac{1}{b-1} \to 2$.

## 3 KEY EXPERIMENTS

We present two key results: (1) small anytime models with SIEVE can outperform large ones with CONST, and (2) EANNs can improve early accuracy, but cost a constant fraction of extra budgets.

**SIEVE vs. CONST of double costs.** In Fig. 2a and Fig. 2b, we compare SIEVE and CONST on ANNs that are based on ResNets on CIFAR100 (Krizhevsky, 2009) and ILSVRC (Russakovsky et al., 2015). The networks with CONST have double the depths as those with SIEVE. We observe that SIEVE leads to the same final error rates as CONST of double the costs, but does so much faster. The two schemes also have similar early performance. Hence, SIEVE effectively speed up the predictions of CONST by about two times. In Fig. 2c, we experiment with the very recent Multi-Scale-DenseNets (MSDNets) (Huang et al., 2017a), which are specifically modified from the recently popular DenseNets (Huang et al., 2017b) to produce the state-of-the-art anytime predictions. We again observe that by improving the final anytime prediction of the smallest MSDNet26 without sacrificing too much early predictions, we make MSDNet26 effectively a sped-up version of MSDNet36 and MSDNet41.

**EANN vs. ANNs and OPT.** In Fig. 2d, we assemble ResNet-ANNs of 45, 81 and 153 conv layers to form EANNs. We compare the EANNs against the parallel OPT, which is from running regular networks of various depths in parallel. We observe that EANNs are able to significantly reduce the early errors of ANNs, but reach the final error rate later. Furthermore, ANNs with more accurate final predictions using SIEVE and EXP-LIN[2] are able to outperform CONST and LINEAR, since whenever an ANN completes in an EANN, the final result is the best one for a long period of time.

---

[2]EXP-LIN is another proposed scheme that focuses more on the final loss. See the full paper for details.

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
