# OpenReview forum: "Anytime Neural Network: a Versatile Trade-off Between Computation and Accuracy"
_ICLR.cc/2018/Conference — Reject_

### Official Review · AnonReviewer3 · 2017-11-27
**Anytime neural network**

**Rating:** 7
**Confidence:** 2

**Review:**

This paper proposes an anytime neural network, which can predict anytime while training. To achieve that, the model includes auxiliary predictions which can make early predictions. Specifically, the paper presents a loss weighting scheme that considers high correlation among nearby predictions, an oscillating loss weighting scheme for further improvement, and an ensemble of anytime neural networks. In the experiments, test error of the proposed model was shown to be comparable to the optimal one at each time budget.

It is an interesting idea to add auxiliary predictions to enable early predictions and the experimental results look promising as they are close to optimal at each time budget.

1. In Section 3.2, there are some discussions on the parallel computations of EANN. The parallel training is not clear to me and it would be great to have more explanation on this with examples.

2. It seems that EANN is not scalable because the depth is increasing exponentially. For example, given 10 machines, the model with the largest depth would have 2^10 layers, which is difficult to train. It would be great to discuss this issue.

3. In the experiments, it would be great to add a few alternatives to be compared for anytime predictions.

---

### Official Review · AnonReviewer1 · 2017-11-28
**Similar to Prior Cascade Work, Unclear Weighing Schemes, and Lack of Experimental Comparison**

**Rating:** 5
**Confidence:** 3

**Review:**

1. Paper Summary

This paper adds a separate network at every layer of a residual network that performs classification. They minimize the loss of every classifier using two proposed weighting schemes. They also ensemble this model.


2. High level paper

The organization of this paper is a bit confusing. Two weighing schemes are introduced in Section 3.1, then the ensemble model is described in Section 3.2, then the weighing schemes are justified in Section 4.1.
Overall this method is essentially an cascade where each cascade classifier is a residual block. Every input is passed through as many stages as possible until the budget is reached. While this model is likely quite useful in industrial settings, I don't think the model itself is wholly original.
The authors have done extensive experiments evaluating their method in different settings. I would have liked to see a comparison with at least one other anytime method. I think it is slightly unfair to say that you are comparing with Xie & Tu, 2015 and Huang et al., 2017 just because they use the CONSTANT weighing schemes.


3. High level technical

I have a few concerns:
- Why does AANN+LINEAR nearly match the accuracy of EANN+SIEVE near 3e9 FLOPS in Figure 4b but EANN+LINEAR does not in Figure 4a? Shouldn't EANN+LINEAR be strictly better than AANN+LINEAR?
- Why do the authors choose these specific weighing schemes? Section 4.1 is devoted to explaining this but it is still unclear to me. They talk about there being correlation between the predictors near the end of the model so they don't want to distribute weight near the final predictors but this general observation doesn't obviously lead to these weighing schemes, they still seem a bit adhoc.

A few other comments:
- Figure 3b seems to contain strictly less information than Figure 4a, I would remove Figure 3b and draw lines showing the speedup you get for one or two accuracy levels.

Questions:
- Section 3.1: "Such an ideal θ* does not exist in general and often does not exist in practice." Why is this the case?
- Section 3.1: " In particular, spreading weights evenly as in (Lee et al., 2015) keeps all i away from their possible respective minimum" Why is this true?
- Section 3.1: "Since we will evaluate near depth b3L/4e, and it
is the center of L/2 low-weight layers, we increase it weight by 1/8." I am completely lost here, why do you do this?


4. Review summary

Ultimately because the model itself resembles previous cascade models, the selected weighings have little justification, and there isn't a comparison with another anytime method, I think this paper isn't yet ready for acceptance at ICLR.

---

### Official Review · AnonReviewer2 · 2017-12-20
**well written paper but lack of justification and comparison**

**Rating:** 5
**Confidence:** 4

**Review:**

This paper aims to endow neural networks the ability to produce anytime prediction. The authors propose several heuristics to reweight and oscillate the loss to improve the anytime performance. In addition, they propose to use a sequence of exponentially deepening anytime neural networks to reduce the performance gap for early classifiers. The proposed approaches are validated on two image classification datasets.
Pros:
- The paper is well written and easy to follow.
- It addresses an interesting problem with reasonable approaches.
Cons:
- The loss reweighting and oscillating schemes appear to be just heuristics. It is not clear what the scientific contributions are.
- I do not fully agree with the explanation given for the “alternating weights”. If the joint loss leads to zero gradient for some weights, then why would you consider it problematic?
- There are few baselines compared in the result section. In addition, the proposed method underperforms the MSDNet (Huang et al., 2017) on ILSVRC2012.
- The EANN is similar to the method used by Adaptive Networks (Bolukbasi et al., 2017), and the baseline “Ensemble of ResNets (varying depth)” in the MSDNet paper.
-  Could you show the error bar In Figure 2(a)? Usually an error difference less than 0.5% on CIFAR-100 is not considered as significant.
- I’m not convinced that AANN really works significantly better than ANN according to the results in Table 1(a). It seems that ANN still outperform AANN in many cases.
- I would suggest to show the results in Table 1(b) with a figure.

---

### Decision · Program_Chairs · 2018-01-29
**ICLR 2018 Conference Acceptance Decision**

**Decision:**

Reject

**Comment:**

The paper received mixed reviews with scores of 5 (R1), 5 (R2),  7 (R3).  All three reviewers raise concerns about the lack of comparisons to other methods. The rebuttal is not compelling on this point. There are quite a few methods that could be used for this application available (often with source code) and should be compared to, e.g. DenseNets (Huang et al.). Given that the proposed method isn't in of itself hugely novel, a thorough experimental evaluation is crucial to the justifying the approach. The AC has closely looked at the rebuttal and the paper and feels that it cannot be accepted for this reason at this time.